# How massed practice improves visual expertise in reading panoramic radiographs in dental students: An eye tracking study

Juliane Richter[1]*, Katharina Scheiter[1,2], Thérése Felicitas Eder[1], Fabian Huettig[3‡], Constanze Keutel[4‡]

1 Leibniz-Institut für Wissensmedien, Tübingen, Germany, 2 Eberhard Karls University of Tübingen, Tübingen, Germany, 3 Department of Prosthodontics, University Hospital for Dentistry, Oral Medicine, and Maxillofacial Surgery at the University Hospital Tübingen, Eberhard Karls University of Tübingen, Tübingen, Germany, 4 Radiology Department of the University Hospital for Dentistry, Oral Medicine, and Maxillofacial Surgery at the University Hospital Tübingen, Eberhard Karls University of Tübingen, Tübingen, Germany

☯ These authors contributed equally to this work.
‡ These authors also contributed equally to this work.
* j.richter@iwm-tuebingen.de

**Data Availability Statement:** All relevant data are within the manuscript and its Supporting Information files.

## Abstract

The interpretation of medical images is an error-prone process that may yield severe consequences for patients. In dental medicine panoramic radiography (OPT) is a frequently used diagnostic procedure. OPTs typically contain multiple, diverse anomalies within one image making the diagnostic process very demanding, rendering students' development of visual expertise a complex task. Radiograph interpretation is typically taught through massed practice; however, it is not known how effective this approach is nor how it changes students' visual inspection of radiographs. Therefore, this study investigated how massed practice–an instructional method that entails massed learning of one type of material–affects processing of OPTs and the development of diagnostic performance. From 2017 to 2018, 47 dental students in their first clinical semester diagnosed 10 OPTs before and after their regular massed practice training, which is embedded in their curriculum. The OPTs contained between 3 to 26 to-be-identified anomalies. During massed practice they diagnosed 100 dental radiographs without receiving corrective feedback. The authors recorded students' eye movements and assessed the number of correctly identified and falsely marked low- and high prevalence anomalies before and after massed practice. Massed practice had a positive effect on detecting anomalies especially with low prevalence ($p < .001$). After massed practice students covered a larger proportion of the OPTs ($p < .001$), which was positively related to the detection of low-prevalence anomalies ($p = .04$). Students also focused longer, more frequently, and earlier on low-prevalence anomalies after massed practice ($p_s < .001$). While massed practice improved visual expertise in dental students with limited prior knowledge, there is still substantial room for improvement. The results suggest integrating massed practice with more deliberate practice, where, for example, corrective feedback is provided, and support is adapted to students' needs.

**Funding:** This research was funded by the Leibniz-WissenschaftsCampus Tübingen "Cognitive Interfaces" (www.wissenschaftscampus-tuebingen.de). The funders had no role in study design, data collection and analysis, decision to publish, or preparation of the manuscript.

**Competing interests:** The authors have declared that no competing interests exist.

# Introduction

Taking radiographs is a standard diagnostic procedure in dentistry. In contrast to other medical disciplines, which rely on the expertise of certified radiologists, dentists perform and interpret radiographs themselves. As in other medical fields, interpretation of medical images is a highly error-prone process in dentistry with error rates between 19% and 41% even for experts [1]. These errors can have severe consequences for patients. Thus, it is crucial that dentistry students develop visual expertise—knowledge about how to search and detect anomalies—during their study [2]. A frequently used and rather traditional instructional method for teaching students how to read and interpret radiographs is massed practice. Here students are required to provide a full written description of their observations including the identified anomalies for each radiograph. This procedure is repeated for a high number of radiographs, which are selected to reflect the full range of potential anomalies that students could be exposed to. No other learning activities are interspersed and only limited corrective feedback, if any, is provided. The reason for the lack of feedback often lies in the fact that medical teachers do not have sufficient time and resources to review their students' diagnostic competence for such a high number of radiographs. Whereas educational research has shown beneficial effects of massed practice for certain types of tasks [3, 4], evidence regarding its effectiveness for the development of visual expertise in medical and dentistry studies is scarce [5]. Moreover, it is yet unclear how students process radiographs, which might have important consequences for their ability to identify anomalies. Therefore, we studied the development of diagnostic competence and gaze behavior in dentistry students during an obligatory standard radiology massed practice course to determine its effects and derive possible implications for improving training.

## Massed practice and the development of visual expertise in radiology

According to Nodine and Mello-Thoms "massed practice is the main change agent in achieving expertise" [6] (p. 868), with a strong relationship between the number of images read and diagnostic accuracy. Moreover, perceptual sensitivity with regard to recognizing low-contrast targets (e.g., lighter areas that may represent nodules) has been shown to improve with massed practice [7]. However, apart from the study of Sowden et al. [7] and anecdotal evidence, to our knowledge there is no evidence that massed practice (without or with corrective feedback) is an effective instructional method for the development of visual expertise.

Nodine and Mello-Thoms [6] delineate that visual expertise requires the development of domain-specific cognitive skills and decision strategies. Observers need to acquire knowledge about perceptual features of anomalies required for their identification. In addition, they need strategies that allow them to interpret conspicuous features by relating them to categories of anomalies. From a theoretical perspective, the mere massed exposure to radiographs without corrective feedback should mainly affect students' ability to match conspicuous features in the images to mental schemata about anomalies (illness scripts, [8]). Massed practice therefore may increase students' experience in finding conspicuous features, which should be evident not only in their ability to correctly identify anomalies (accuracy) but also in their gaze behavior as a fine-grained process-oriented measure of visual expertise. Eye tracking serves as a valuable research tool to study visual expertise development in the medical field [2, 5, 9]. According to this research, experts tend to fixate images for a shorter time and have more and earlier fixations on relevant areas containing conspicuous features compared to non-experts [10]. During a fixation the eyes remain relatively still on a certain location, which allows information intake or active processing of perceptual features [5, 11]. Importantly, while there is a wealth of studies addressing expert-novice differences in medical image interpretation [12–

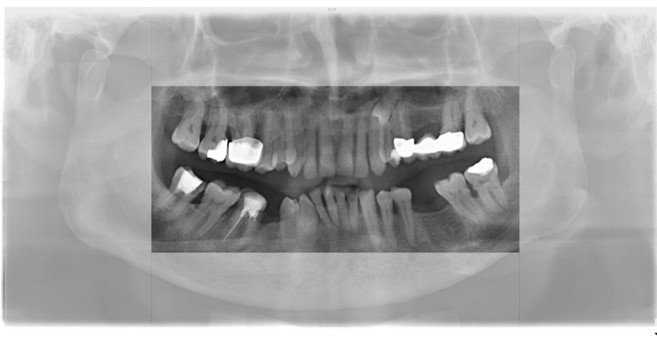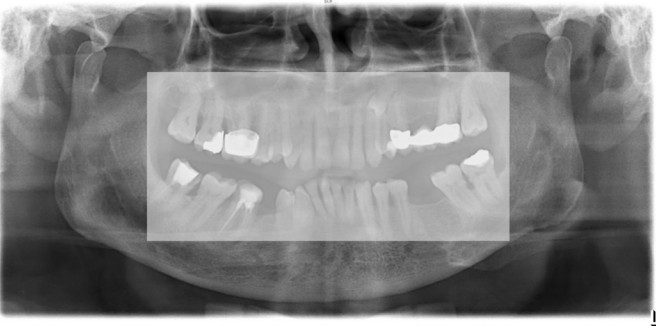

**Fig 1. Example orthopantomogram (OPT).** The left panel shows an OPT with a highlight on the central area whereas the right panel highlights the peripheral area of an OPT.

14], there are no studies to the authors' knowledge on how visual expertise develops during unversity training of students.

Moroever, there is hardly any research (two studies see [15, 16]) on medical image interpretation regarding panoramic radiographs (orthopantomograms, OPTs), which are frequently used in dentistry, and which is our area of interest. OPTs typically contain multiple, diverse anomalies within one image [17]. Typical anomalies of the dentition are located in the central area of an OPT (Fig 1) around the teeth and adjacent alveolar bone, for example, root remnants, periodontal defects, and apical lesions. The peripheral area of an OPT (Fig 1) shows the temporomanibular joints, maxillary sinus, parts of the orbital cavities, and soft tissues of the neck including the hyoid bone, where typical anomalies contain cysts or tumors of the soft and hard tissue or calcifications of salivary glands, lymph nodes or the carotid artery. Anomalies located in the peripheral area are of rather low prevalence [18, 19] whereas high prevalence anomalies are predominantly located in the central area.

Importantly, the task of diagnosing an OPT is very different from diagnosing, for example, mammograms or chest radiographs that have been used in previous studies on visual expertise in radiology, e.g., [14, 20–26]. For example, in a study by Donovan, Manning and Crawford [27] chest radiographs were presented to participants with the task to detect lung nodules only. In a mammography screening-diagnostic task, Nodine and colleagues [26] asked experts with different levels of experience to detect malignant lesions; Mammography radiographs typically contain only a modest number of lesions, different from OPTs that can entail a large number of very different anomalies. Consequently, it may be particularly challenging to develop visual expertise regarding OPTs for a number of reasons:

A high interindividual variability of the visual appearance of both normal and abnormal anatomy as well as the phenomena caused by artifacts and superimpositions makes it difficult to detect anomalies [28–31]. In addition, even panoramic radiographs of apparently "dentally healthy patients" can often show one or more dental or non-dental anomalies [32–36]. Therefore, diagnosing OPTs relies on hybrid search for multiple different targets requiring observers to know all characteristics of potential targets, and match those to actual visual characteristics of radiographs [37]. The occurrence of multiple targets is known to complicate visual search and make it less effective; the prevalence of targets also affects visual search processes [38]. Moreover, anomalies are also found in the peripheral areas of the jawbone or in the maxillary sinus or are part of the X-ray as superimpositions of soft and hard tissue in the vicinity of the oral cavity. These may include a number of secondary findings of general medical relevance (oncology, cardiovascular disease) that require referral to other specialists for further diagnosis. To

conclude, there are several reasons why studies in other medical imaging domains have limited informative value for teaching and learning the task of diagnosing OPTs [39].

Against the backdrop of the rather weak empirical basis regarding the effectiveness of massed practice for the development of visual expertise, we investigated the following research questions:

1. Does massed practice improve diagnostic accuracy when reading OPTs and does its effectiveness depend on prevalence rates and hence the anomaly's location in either the center or the periphery?

2. How does massed practice change students' gaze behavior when reading OPTs as revealed by tracking students' eyes during the inspection of OPTs? Based on visual expertise research, we expected that an increase in diagnostic performance due to massed practice should be accompanied by earlier, longer, and more frequent fixations on anomalies. Moreover, we expected students to cover a larger proportion of an OPT during visual search after massed practice.

In brief, our results show that as expected massed practice had a positive effect on detecting anomalies, especially for low-prevalence anomalies. Also, we provided evidence that massed practice leads to changes in gaze behavior. After massed practice students covered a larger proportion of the OPTs during visual inspection and their coverage positively predicted the detection of low-prevalence anomalies. Students also focused longer, more frequently, and earlier on low-prevalence anomalies after massed practice.

## Materials and methods

### Participants

Sixty-nine dental students participated in this study. They were tested three times during their first clinical semester, which is the second half of their third year: (i) prior to massed practice (pre-test), (ii) directly after massed practice (post-test), and (iii) at the end of the semester (13 weeks after pre-test). Because the analyses showed no differences in dependent variables between the second and third measurement, we decided to use only the pre- and post-test for the analyses. In cases for which the second measurement was missing, we replaced it with values from the third measurement ($n$ = 5 students). Nevertheless, data from 14 students had to be excluded due to incompleteness, leaving 55 students ($M_{age}$ = 24.05, SD = 2.56 years old; 61.8% female). Due to insufficient eye tracking quality and/or calibration accuracy, data from another eight students had to be excluded from the eye-tracking analyses. The remaining 47 students were $M_{age}$ = 23.94, SD = 2.51 years old and 59.6% were female. The study was approved by the Ethical Review Board of the Leibniz-Institut für Wissensmedien Tübingen under number LEK 2017/016. Participation in the study was voluntary; all students provided written informed consent including that their data could be analyzed and published.

### Materials and apparatus

Students were asked to mark anomalies in 10 OPTs that had been taken during routine diagnostic processes in the hospital and were of good image quality. Those 10 OPTs showed between three and 26 anomalies ($k$ = 95 anomalies in total). No normal images (radiographs without pathological findings) were included. We used the same ten OPTs in the pre- and post-tests because research suggests that observers do not recognize previously seen radiographs, which suggests that the repetition of OPTs in the pre and post-test may only have little if any effect on diagnostic performance [25, 40, 41].

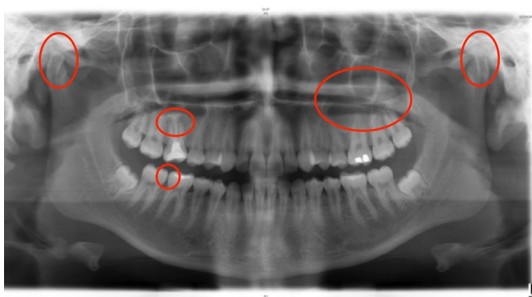 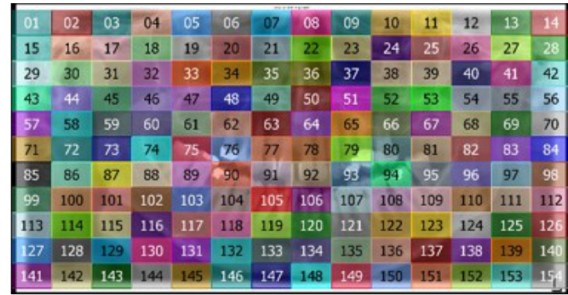

**Fig 2. Example orthopantomogram (OPT) with areas of interest (AOIs).** The left panel shows AOIs located around anomalies on an OPT. In addition to a bilateral shortening of the collum mandibulae, the colla and condyles present hypoplastic on both sides. Shortened roots of tooth 16, lacking the apical tips with periapical translucencies, possibly corresponding to a status post root resection. In region 26, 27 and 28 spheroid, sharply defined homogeneous opacification (projecting on the maxillary sinus floor) corresponding to a mucosal antral pseudocyst. Translucencies in the approximal areas of teeth 46 and 47 possibly indicating caries. (Dental notation according to the FDI-system). Gridded AOIs, as displayed in the right panel were used for the computation of the overall gaze coverage.

Two experts (a maxillofacial radiologist and a prosthodontist, each with over 13 years of clinical experience) selected and coded the OPTs independently and agreed on a solution scheme for coding the students' responses.

Stimuli were presented on a computer screen (1920 x 1080 pixels) and at maximum screen brightness. Eye movements were recorded using a video-based remote eye tracking system by SensoMotoric Instruments™ (SMI 250RED™; 250 Hz sampling rate). A 13-point calibration image was used to calibrate the system. We used the SMI BeGaze™ default velocity-based algorithm (eye movements with a speed lower than 40°/s were classified as fixations; eye movements with a speed above 40°/s as saccades) to detect events in the gaze data. The calibration accuracy was below 0.98° visual angle. The mean tracking ratio was 95.03% at the first and 94.92% at the second measurement. The light in the experimental room was kept constant throughout the experiment (range: 30 to 40 lx).

We aggregated gaze data by areas of interest (AOIs) in two different ways: (i) we drew AOIs around anomalies to compute fixation time, fixation count, and the time to first fixating anomalies, and (ii) we used gridded AOIs (14x11 and 15x11 grids depending on the size of the OPT) to compute the overall gaze coverage of the OPTs (see Fig 2).

### Instruments

**Accuracy.** We computed the accuracy as the sum of correctly identified anomalies separately for the central and peripheral parts, respectively (see Fig 1) of the OPTs and transformed theses scores into percentages for easier interpretation.

**False positives.** When students marked an area in an OPT as being abnormal but did not actually contain an anomaly, we coded those markings as false positives, with reference to either the central or peripheral area of the OPTs.

**Eye tracking parameters.** We used four measures related to students' gaze behavior: (i) the mean fixation time on central and peripheral anomalies (in milliseconds), (ii) mean number of fixations on central and peripheral anomalies, (iii) mean time to first fixation on central and peripheral anomalies (in milliseconds), and (iv) the coverage of OPTs as the percentage of grids that were fixated at least once. The first fixation on each OPT was excluded from data analyses, because this fixation can be traced back to the fixation cross, that was presented just before each OPT. Data were averaged across stimuli.

**Dental pathology test.** A multiple-choice test with 20 items assessed knowledge about dental pathology (e.g., misaligned teeth, root resorptions, soft tissue issues). The items were

self-developed with each five answer options (including 'I cannot answer the question yet/I don't know') and one option being correct.

## Experimental procedure

We collected the data in multiple group sessions. In the pre-test session, students were asked to perform a diagnostic task, which was to first look at the OPT (limited to 90 seconds) and then mark anomalies (without time constraint). In the marking phase students used a mouse-operated drawing tool to draw ellipses around conspicuous areas. Before each OPT, a fixation cross was displayed for two seconds. After students had diagnosed all 10 OPTs, they worked on the dental pathology test. In the remaining semester students attended weekly lectures on radiology. The lectures addressed radiation physics and biology, radiation exposure and protection, dosimetry, technical equipment, imaging procedures, quality control, legal directives, and technical exercises. In addition, OPTs are introduced including the clarification of anatomical structures, common anomalies, and artefacts from technical failures as well as common anomalies. Finally, students perform massed practice of reading dental radiographs. Within 24 hours spread across one week each student diagnosed 100 dental radiographs with a written report for each. As to expect due to prevalence 15–20% were without pathological findings. The students worked in teams of three without receiving any corrective feedback by the teacher. Thereafter, two out of these hundred radiographs are discussed in depth with each student together with an experienced radiologist (teacher). At the end of the massed practice week, students were invited to the post-test session. In the post-test we asked them to repeat the diagnostic task used in the pre-test. The diagnostic task was repeated at the end of the semester; in addition, the dental pathology test was administered again.

## Statistical analyses

Repeated-measures analyses of variance (ANOVA) with two within-subjects factors time of measurement (ToM; pre/post massed practice) and anomaly location (AL; central/peripheral area of an OPT) were used to determine the effects of massed practice on accuracy, false positives, fixation time, number of fixations, and time to first fixation. We used Bonferroni-corrected post-hoc comparisons to disentangle significant interactions. For the gaze coverage of OPTs and the control variable dental pathology knowledge we computed repeated-measures ANOVAs only with the within-subjects factor ToM. Finally, a correlation analysis was conducted to test how changes in gaze behavior were related to changes in accuracy. For the ANOVAs effect sizes are reported in $\eta_p^2$ to denote small (range from 0.01 to 0.05), medium (range from 0.06 to 0.13), or large effects (from 0.14 upwards), respectively. The effect size *d* is used to denote small (range from .20 to .40), medium (range from .50 to .70) and large effects (from .80 upwards) resulting from pairwise comparisons [42]. The alpha level was set to .05.

## Results

### Dental pathology knowledge

Students' knowledge increased significantly from the beginning to the end of the semester, $F(1,48) = 151.45$, $p < .001$, $\eta_p^2 = .76$ (Table 1). However, these knowledge gains were unrelated to diagnostic accuracy (r = -.07, $p = .639$) and gaze coverage after massed practice (r = .05, $p = .748$).

### Diagnostic competence

**Accuracy.** Accuracy improved after massed practice, $F(1,54) = 431.10$, $p < .001$, $\eta_p^2 = .89$, and differed between AL, $F(1,54) = 72.28$, $p < .001$, $\eta_p^2 = .57$. These main effects were qualified

**Table 1. Means and standard deviations for dental pathology knowledge as a function of the time of measurement.**

| ToM | Dental pathology knowledge in % correct (standard deviation) |
|---|---|
| Pre* | 18.98 (3.58) |
| Post** | 41.33 (12.28) |

SD, standard deviation; ToM, time of measurement.

* Cronbach's alpha 0.82.

**Cronbach's alpha 0.51.

n = 49.

by a significant interaction, $F(1,54) = 49.86$, $p < .001$, $\eta_p^2 = .48$: Accuracy increased for both central and peripheral anomalies from pre- to the post-test (both $p_s < .001$), but this increase was stronger for peripheral (d = 2.91) than for central anomalies (d = 1.52) (Table 2).

**False positives.** Because the number of false positives was not normally distributed, we log-transformed the variables (Table 2). Results revealed main effects for ToM, $F(1,54) = 45.46$, $p < .001$, $\eta_p^2 = .46$, and AL, $F(1,54) = 251.31$, $p < .001$, $\eta_p^2 = .82$, as well as a marginally significant interaction, $F(1,54) = 3.65$, $p = .062$, $\eta_p^2 = .06$. The number of false positives in the central areas of OPTs did not change ($p = .524$), whereas it increased significantly from before to after massed practice in the peripheral area ($p < .001$). Fig 3 shows students' diagnostic competence reflected in the accuracy for finding anomalies and the number of falsely marked anomalies as a function of ToM and AL.

## Gaze behavior

**Fixation time.** There were main effects of ToM, $F(1,46) = 18.14$, $p < .001$, $\eta_p^2 = .28$, and AL, $F(1,46) = 102.48$, $p < .001$, $\eta_p^2 = .69$, as well as an interaction, $F(1,46) = 93.24$, $p < .001$, $\eta_p^2 = .67$, for the fixation time on anomalies. Whereas fixations on central anomalies were shorter after massed practice ($p < .001$), fixation times for peripheral anomalies increased from pre- to post-test ($p < .001$) (Table 3).

**Table 2. Means and standard deviations for diagnostic performance as a function of anomaly location and time of measurement.**

| ToM and AL | Percentage of correctly detected anomalies♭ (standard deviation) [raw scores] | Number of false positives (standard deviation) [log-transformed values♭] |
|---|---|---|
| Pre | | |
| Central anomalies | 35.27 (10.30) | 14.33 (11.74) |
|  | [27.51 (8.04)] | [1.09 (0.29)] |
| Peripheral anomalies | 12.41 (11.75) | 1.04 (1.52) |
|  | [2.11 (2.00)] | [0.22 (0.27)] |
| Post | | |
| Central anomalies | 51.40 (10.97) | 15.51 (12.03) |
|  | [40.09 (8.56)] | [1.12 (0.30)] |
| Peripheral anomalies | 48.34 (12.94) | 6.11 (5.65) |
|  | [8.22 (2.20)] | [0.71 (0.38)] |

ToM, time of measurement; AL, anomaly location.

♭Used for the analyses.

n = 55.

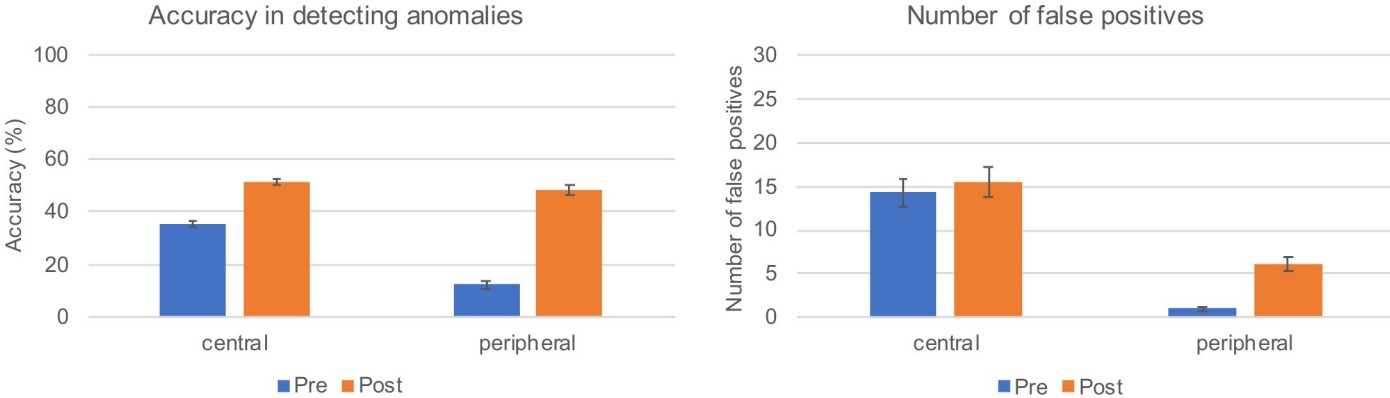

**Fig 3. Students' diagnostic performance as a function of ToM (pre and post massed practice) and AL (central/peripheral).** The left panel depicts the accuracy in detecting anomalies whereas the right panel shows the number of falsely marked anomalies.

**Number of fixations.** An analogous pattern holds true for the number of fixations (Table 3). There were main effects of ToM, $F(1,46) = 14.91$, $p < .001$, $\eta_p^2 = .25$, and AL, $F(1,46) = 249.01$, $p < .001$, $\eta_p^2 = .84$, as well as a significant interaction, $F(1,46) = 85.68$, $p < .001$, $\eta_p^2 = .65$. From pre- to post-test the number of fixations significantly decreased for central anomalies ($p < .001$) but increased for peripheral anomalies ($p < .001$).

**Time to first fixation.** Results for the time to first fixating anomalies revealed a significant main effect only for AL, $F(1,46) = 31.57$, $p < .001$, $\eta_p^2 = .41$; ToM: $F < 1$. Moreover, there was a significant interaction of ToM and AL, $F(1,46) = 44.30$, $p < .001$, $\eta_p^2 = .49$. After massed practice central anomalies were fixated later ($p < .001$), whereas peripheral anomalies were fixated earlier ($p < .001$) compared to before massed practice (Table 3). Fig 4 depicts the number of fixations on anomalies and the time to first fixating anomalies as a function of ToM and AL.

**Gaze coverage.** The coverage of OPTs increased significantly after massed practice, $F(1,46) = 274.69$, $p < .001$, $\eta_p^2 = .86$ (see Table 3).

## Correlations among change scores for accuracy and gaze behavior

We computed change scores for accuracy and gaze behavior measures by subtracting each pre-test value from the value achieved in the post-test. A correlation analysis showed that the increase in accuracy for central anomalies was not related to any changes in gaze behavior. The increase in the accuracy for peripheral anomalies was, however, positively correlated with an increase in fixation time ($p = .037$) and fixation count ($p = .017$) on peripheral anomalies and with an increase in gaze coverage ($p = .035$) (Table 4).

**Table 3. Means and standard deviations (in parentheses) for variables related to gaze behavior as a function of anomaly location and the time of measurement.**

| ToM and AL | Fixation time (ms) | Number of fixations | Time to first fixation (ms) | Coverage (percentage) |
|---|---|---|---|---|
| Pre | | | | 32.11 (4.70) |
| Central anomalies | 2,421.77 (741.17) | 4.05 (1.22) | 23,310.76 (4,998.91) | |
| Peripheral anomalies | 2,436.31 (933.24) | 5.55 (2.19) | 23,960.42 (7,788.68) | |
| Post | | | | 45.59 (4.93) |
| Central anomalies | 1,662.70 (541.19) | 2.89 (0.71) | 28,462.83 (5,758.25) | |
| Peripheral anomalies | 4,139.66 (1,306.08) | 8.81 (2.34) | 17,140.60 (6,243.09) | |

ToM, time of measurement; AL, anomaly location.

n = 47.

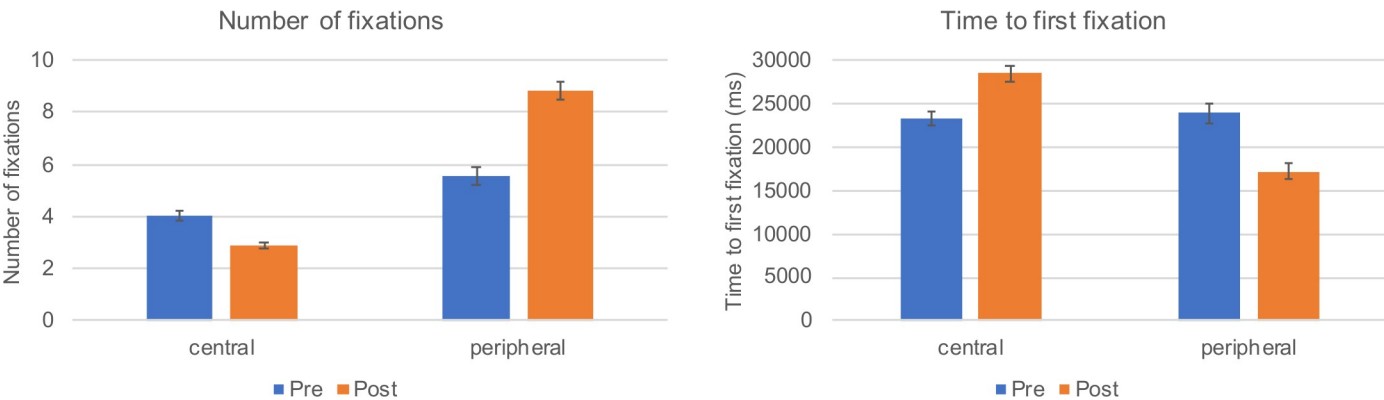

**Fig 4. Students' gaze behavior as a function of ToM (pre and post massed practice) and AL (central/peripheral).** The left panel depicts the number of fixations on anomalies whereas the right panel shows the time to first fixating anomalies.

## Discussion

We investigated the effectiveness of massed practice for learning how to read and interpret panoramic radiographs. To this end, we assessed diagnostic performance and gaze behavior before and after a regular massed practice university course. Results revealed that massed practice is an effective instructional method for improving students' accuracy in detecting anomalies and training them to not only focus on central areas, but also frequently neglected peripheral areas. These improvements were positively related with more attention being paid to these areas, indicating that students modify their visual search strategies due to massed practice. In addition, the improved visual coverage after massed practice positively predicted the accuracy of detecting anomalies in the periphery [22]. These results are promising because they demonstrate the effectiveness of massed practice for learning how to perform the complex hybrid search task of diagnosing OPTs containing multiple, diverse anomalies. Moreover, our study showed that regular massed practice training with 100 radiographs already significantly changes students' viewing behavior with more and longer fixations on relevant areas of an OPT [10].

Despite revealing improvements, our results also reveal limitations of massed practice as an instructional method. Students made more false positive markings in the periphery after massed practice. Massed practice trained students in finding and matching conspicuous features in the images to their mental schemata about anomalies. Since students adapted their visual search strategies and more fully covered OPTs, it seems that students' mental schemata

**Table 4. Correlations among change score values for accuracy and gaze behavior measures.**

| AL | Change scores post-pre | Accuracy central anomalies | Accuracy peripheral anomalies |
|---|---|---|---|
| central | Fixation time | .19 | |
| | Fixation count | -.09 | |
| | Time to first fixation | .02 | |
| peripheral | Fixation time | -.19 | .31* |
| | Fixation count | -.22 | .35* |
| | Time to first fixation | .21 | -.11 |
| Independent of AL | Gaze coverage | -.27 | .31* |

AL, anomaly location.

* $p < .05$.

** $p < .01$. $n = 47$.

about the visual features of anomalies could be further improved by means of systematic and repeated training addressing the variability in the visual appearance of anomalies.

Moreover, students' accuracy was at about 50% after massed practice, which leaves substantial room for improvement. From our cross-sectional data collection of all dentistry semesters we know that this accuracy level does not change during the further course of their studies. This is despite the fact that students gain further experience by treating their own patients and interpreting radiographs together with supervising experienced dentists [43]. Potential reasons for the stagnation of accuracy may be that this learning process is not systematically oriented towards diagnosing and interpreting radiographs and is strongly influenced by the patient cases available for treatment and the focus of the supervising dentist. Both findings suggest combining massed practice with more deliberate practice [44, 45]. According to the deliberate practice approach domain-specific expertise is the result of structured practice. It is characterized by the adaptation of contents to learners' expertise level and repetition of contents. Individualized feedback is an important aspect of deliberate practice because it draws attention to those aspects of one's performance that need correction and further practice [46]. Moreover, research consistently shows that spaced practice leads to better learning outcomes compared to massed practice for varying types of tasks (e.g., verbal memory tasks, motor learning) [47–49]. Spaced practice means that the contents to be learned are repeated after a certain time interval and are tested after a further retention interval [47, 50]. Also in radiology teaching and surgical skill training, studies have shown that spaced procedures are beneficial compared to massed practice [51, 52]. Future research should therefore specifically compare massed and spaced practice for learning how to diagnose OPTs to identify potentials for further improvements of students' accuracy. The present study did not aim at contrasting different training approaches because our focus was on the effects that current practice has on students' skill development. Therefore, we implemented the present study in a within-subjects design within students' regular training to achieve a high ecological validity. We acknowledge that this approach has its limitations due to a lack of a control group that would have been trained with a different approach.

Importantly, the current study refers only to the effects of massed practice regarding training of a *single* skill, where it has been suggested that training of that single skill should occur spaced in time. This recommendation is not to be confused with another instructional design principle, for which multiple labels have been used in the literature, that is, the contextual inference effect [53–55], the variability effect [56, 57] or interleaved practice [58], respectively. Here it is suggested that when training *multiple* skills, these should be trained in an interleaved way (abcab-cabc etc.) to highlight the differences between them. While interleaved practice of multiple skills results in spaced learning of the same skills, it is different from the focus of the present study.

To summarize, the present study complements existing research on medical image interpretation [12–14, 59–61] in that it focuses on the effects of training of students rather than on expert-novice differences and it uses OPTs rather than, for instance, chest radiographs, which require different types of visual search processes. Our results indicate that traditional massed practice training is an effective instructional method to develop visual expertise for interpreting OPTs in dentistry students. Students do not only improve their diagnostic accuracy but also change their visual search behavior due to massed practice training. However, at the same time the effectiveness of massed practice is limited. Further improvements may be achieved by combining massed practice with more systematic training such as deliberate practice.

## Supporting information

**S1 Data. This is the data file used for the analyses reported in the present manuscript.** (SAV)

## Acknowledgments

We thank the dental students' representatives (Fachschaft Zahnmedizin Tübingen) for contributing in motivating students to participate in the study.

## Author Contributions

**Conceptualization:** Juliane Richter, Katharina Scheiter, Fabian Huettig, Constanze Keutel.

**Data curation:** Juliane Richter, Thérése Felicitas Eder.

**Formal analysis:** Juliane Richter.

**Funding acquisition:** Katharina Scheiter, Fabian Huettig, Constanze Keutel.

**Investigation:** Juliane Richter, Thérése Felicitas Eder.

**Methodology:** Juliane Richter, Katharina Scheiter.

**Project administration:** Katharina Scheiter, Fabian Huettig, Constanze Keutel.

**Supervision:** Katharina Scheiter, Fabian Huettig, Constanze Keutel.

**Validation:** Juliane Richter.

**Visualization:** Juliane Richter, Thérése Felicitas Eder.

**Writing – original draft:** Juliane Richter.

**Writing – review & editing:** Juliane Richter, Katharina Scheiter, Thérése Felicitas Eder, Fabian Huettig, Constanze Keutel.

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
