## [Decision Letter · Decision Letter 0]

28 Jul 2020

PONE-D-20-05443

How massed practice improves visual expertise in reading panoramic radiographs in dental students: An eye tracking study

PLOS ONE

Dear Dr. Richter,

Thank you for submitting your manuscript to PLOS ONE. After careful consideration, we feel that it has merit but does not fully meet PLOS ONE’s publication criteria as it currently stands. Therefore, we invite you to submit a revised version of the manuscript that addresses the points raised during the review process.

We look forward to receiving your revised manuscript.

Kind regards,

Ezio Lanza, M.D.

Academic Editor

PLOS ONE

Journal Requirements:

2. We noted in your submission details that a portion of your manuscript may have been presented or published elsewhere.

"At this point in time one other manuscript is under review that uses data from the same student population we used in the present study. However, they only used the measure pupil diameter, which is not included in our present manuscript. Therefore, there are no overlaps between the two manuscripts regarding results, data, or figures. But if the reviewers feel that they want to check the overlap themselves we are happy to share the other manuscript upon request."

Please clarify whether this publication was peer-reviewed and formally published. If this work was previously peer-reviewed and published, in the cover letter please provide the reason that this work does not constitute dual publication and should be included in the current manuscript.

5. Review Comments to the Author 

Reviewer #1: The authors are to be commended for their innovative work, as they sought to “assess the development of diagnostic competence and gaze behavior in dentistry students during an obligatory standard radiology massed practice course”.

Nonetheless, as it is, the manuscript presents two major and important weaknesses 

a) The design of the study does not include a control group that might allow us effectively assess and compare different learning outcomes. Other studies using a spaced learning/distributed practice approach appear to show that the use of spacing and testing promotes long-term or durable memory (Morin CE et al. Pediatr Radiol. 2019; 49: 990-999 and Versteeg M et al. Med Educ. 2020; 54: 205-216);

b) Moreover, despite the presented data, there is little to no discussion of the impact on the actual learning outcomes of the students who participated in the study. If one is to assess massed practice, one should also discuss and comment on alternative approaches such as distributed practice.

Unless the authors can, somehow, address these key aspects, 

Reviewer #2: Interesting article with new information. Here are a few suggestions/recommendations

- Line 110: I'd suggest using "hyoid bone" instead of only "hyoid)

- Line 131: You mention a few studies (# 15-16 and 20 to 26) but do not provide explanation as to what these studies brought. If pertinent, maybe you could expand on at least a few of these

- Line 132: "oral sinus"... are you talking about the maxillary sinus?

- Line 135: "for why". I think the "for" is not needed here

- Line 146: you talk about "increases", plural. But I'm not sure to what multiple things you are referring to

- Participant section: I'd like to know what year were these students in?

- Methods section: you mention the 1st test, 2nd test, and then a 3rd test 13 weeks after the 1st. When was the 2nd test administered?

- Lines 185 and227/228: I don't think it's relevant to indicate your experts as the authors. Simply explaining their background should be enough.

- Figure 2: I'm not sure I see what's wrong with the TMJs in this case? It might be worth explaning in you caption (lines 204-206) what the experts saw on this image.

Reviewer #3: The article is simple but the idea is satisfactory.

About English language: The manuscript needs an English review in order to sound as native as possible, preferably changing some words that can be understood but are not routine words in academic writing. For example, in the first phrase of the introduction, the words: "Taking radiographs" sound weird. Also, in academic writing, passive voice is preferable, and the article is almost entirely wrote in active voice.

Also, punctuation needs review.

# Introduction:

Please, provide a reference to the phrase: "The reason for the lack of feedback often lies in the fact that medical teachers do not have sufficient time and resources to review their students’ diagnostic competence for such a high number of radiographs." - I don't think the introduction is the right place to author's opinion.

I suggest you to explain further the meaning of "massed practice" with proper references. Explain how the practice is performed in a summarized and systematic explanation. Maybe you can use a figure or a flow chart to illustrate and clearer this point.

In the phrase: "According to Nodine and Mello-Thoms [6]“massed practice is the main change

76 agent in achieving expertise” (p. 868)," please, check if the reference citation is correct.

Please, add references to the phrases: "As a consequence, it may be

122 particularly challenging to develop visual expertise regarding OPTs for a number of

123 reasons. A high interindividual variability of the visual appearance of both normal and

124 abnormal anatomy makes it difficult to detect anomalies. In addition, even OPTs of

125 apparently healthy patients can often show several anomalies"

# Discussion

I suggest you add more references in order to compare your findings. For instance:

Again, maybe it would be better to express your opinion in the discussion, for exemple. Remove or re-wright theses paragraphs.

Introduction, last two paragraphs: I don't think Introduction is the right place to express your opinion and what you expect from the study. You can express it in the discussion.

# Materials and Methods

Participants: you mentioned 55 students in the abstract, however, in the first paragraph of M&M you mentioned, after exclusion criteria applied, 47 students as your final sample. Please, correct it or clarify the numbers divergence.

#Discussion

I suggest you to improve your discussion with other references and comparisons with different studies, such as, for example:

* Performance evaluation of different observers in the interpretation of panoramic radiographs by the mandibular cortical index. DOI: 10.15448/1980-6523.2018.1.29202

*Observer performance in diagnosing osteoporosis by dental panoramic radiographs: results from the osteoporosis screening project in dentistry: https://doi.org/10.1016/j.bone.2008.03.014

---

## [Author Response · Author response to Decision Letter 0]

1 Oct 2020

Dear Dr. Lanza, dear Reviewers, 

we would like to thank the editor and the reviewers for the assessment of our manuscript and the feedback offered by the review. In the following, we will address the comments one by one. Reviewer comments are printed in regular font, responses in italics. Moreover, we have added sections from the manuscript whenever we made a change in it (in italics and quotation marks). We hope that this way we have addressed all issues raised by the reviewers and the editor in a satisfactory manner. 

The authors 

5. Review Comments to the Author 

Reviewer #1: 

The authors are to be commended for their innovative work, as they sought to “assess the development of diagnostic competence and gaze behavior in dentistry students during an obligatory standard radiology massed practice course”.

Thank you for your time to review our manuscript and providing constructive feedback.

Nonetheless, as it is, the manuscript presents two major and important weaknesses 

a) The design of the study does not include a control group that might allow us effectively assess and compare different learning outcomes. Other studies using a spaced learning/distributed practice approach appear to show that the use of spacing and testing promotes long-term or durable memory (Morin CE et al. Pediatr Radiol. 2019; 49: 990-999 and Versteeg M et al. Med Educ. 2020; 54: 205-216);

Thank you for your comment. Indeed, we did not use a control group because we did not aim at contrasting different training approaches. Not only would this have been difficult to implement from an ethical and practical perspective, since we would have experimentally manipulated students’ real study conditions that may affect passing/failing the course. Moreover, our focus was more on the effects that current practice has on students’ skill development. Therefore, we felt that it was necessary to implement our study within students’ regular training to achieve a high ecological validity. Hence, from an experimental psychologist’s point of view, massed practice would have to serve as our experimental group, but it is unclear what the control group would be. A waiting-control group cannot be implemented for practical reasons. Hence, we favored a within-subjects design where the pretest represents the performance level of somebody who has received no training. However, we agree that future research should focus on introducing and evaluating learning approaches that from a psychology perspective can be expected to result in better performance such as deliberate practice and distributed/spaced learning. We added the potential use of spaced learning respectively distributed practice in the discussion section:

Line 481

“Moreover, research consistently shows that spaced practice leads to better learning outcomes compared to massed practice for varying types of tasks (e.g., verbal memory tasks, motor learning) [47–49]. Spaced practice means that the contents to be learned are repeated after a certain time interval and are tested after a further retention interval [47,50]. Also, in radiology teaching, studies have shown that spaced procedures are beneficial compared to massed practice [51,52].”

(added) References:

47. Cepeda NJ, Vul E, Rohrer D, Wixted JT, Pashler H. Spacing effects in learning: A temporal ridgeline of optimal retention: Research article. Psychol Sci. 2008;19: 1095–1102. doi:10.1111/j.1467-9280.2008.02209.x

48. Cepeda NJ, Pashler H, Vul E, Wixted JT, Rohrer D. Distributed practice in verbal recall tasks: A review and quantitative synthesis. Psychol Bull. 2006;132: 354–380. doi:10.1037/0033-2909.132.3.354

49. Logan JM, Castel AD, Haber S, Viehman EJ. Metacognition and the spacing effect: The role of repetition, feedback, and instruction on judgments of learning for massed and spaced rehearsal. Metacognition Learn. 2012;7: 175–195. doi:10.1007/s11409-012-9090-3

50. Versteeg M, Hendriks RA, Thomas A, Ommering BWC, Steendijk P. Conceptualising spaced learning in health professions education: A scoping review. Med Educ. 2020;54: 205–216. doi:10.1111/medu.14025

51. Morin CE, Hostetter JM, Jeudy J, Kim WG, McCabe JA, Merrow AC, et al. Spaced radiology: encouraging durable memory using spaced testing in pediatric radiology. Pediatr Radiol. 2019;49: 990–999. doi:10.1007/s00247-019-04415-3

52. Rozenshtein A, Pearson GDN, Yan SX, Liu AZ, Toy D. Effect of massed versus interleaved teaching method on performance of students in radiology. J Am Coll Radiol. 2016;13: 979–984. doi:10.1016/j.jacr.2016.03.031

b) Moreover, despite the presented data, there is little to no discussion of the impact on the actual learning outcomes of the students who participated in the study. If one is to assess massed practice, one should also discuss and comment on alternative approaches such as distributed practice.

Thank you for pointing out the necessity of discussing the impact of massed practice on learning outcomes more thoroughly and discussing further alternative learning approaches other than deliberate practice. We also think these are important aspects, which is why we expanded the discussion section respectively as we explained in the previous comment. Nevertheless, it is important to note that despite the fact that there may be more effective approaches to teaching image interpretation skills from a psychology perspective, massed practice is the favored approach in medical education practice. Thus, our study serves to make a statement of what can be achieved using this approach, while at the same time discussing possible alternatives that seem recommendable from a psychology perspective. Hence, the purpose of the paper was not to identify the most optimal training approach, but rather to study the impact of current practice. 

Unless the authors can, somehow, address these key aspects, 

Unfortunately, this statement is incomplete. 

Reviewer #2: 

Interesting article with new information. Here are a few suggestions/recommendations

Thank you for this overall positive assessment of our manuscript and the constructive feedback.

- Line 110: I'd suggest using "hyoid bone" instead of only "hyoid)

Thank you. We changed the term.

- Line 131: You mention a few studies (# 15-16 and 20 to 26) but do not provide explanation as to what these studies brought. If pertinent, maybe you could expand on at least a few of these

Thank you for this comment. We agree that it may be useful for readers to get some more information about the studies that investigated diagnosing tasks for radiographs other than OPTs. 

Line 143

“For example, in a study by Donovan, Manning and Crawford [27] chest radiographs were presented to participants with the task to detect lung nodules only. In a mammography screening-diagnostic task, Nodine and colleagues [26] asked experts with different levels of experience to detect malignant lesions; Mammography radiographs typically contain only a modest number of lesions, different from OPTs that can entail a large number of very different anomalies. Consequently, it may be particularly challenging to develop visual expertise regarding OPTs for a number of reasons. A high interindividual variability of the visual appearance of both normal and abnormal anatomy as well as the phenomena caused by artifacts and superimpositions makes it difficult to detect anomalies [28–31]. In addition, even panoramic radiographs of apparently “dentally healthy patients” can often show one or more dental or non-dental anomalies [32–36].

- Line 132: "oral sinus"... are you talking about the maxillary sinus?

Indeed, what we were referring to is the oral cavity. We changed the term accordingly (Line 161).

- Line 135: "for why". I think the "for" is not needed here

Changed.

- Line 146: you talk about "increases", plural. But I'm not sure to what multiple things you are referring to

It is an increase in diagnostic performance, which is singular as you noticed correctly. We changed it accordingly. Thank you.

- Participant section: I'd like to know what year were these students in?

Students in their first clinical semester were in the second half of their third year. We added this information in line 199.

- Methods section: you mention the 1st test, 2nd test, and then a 3rd test 13 weeks after the 1st. When was the 2nd test administered?

The second test was administered directly after massed practice (see line 200) and therefore depended on the date when students finished their massed practice training. They were trained in small groups consisting of 2 to 4 students, which is why massed practice training was distributed across the semester in order to train all 69 students. 

- Lines 185 and227/228: I don't think it's relevant to indicate your experts as the authors. Simply explaining their background should be enough.

Thank you, we changed it.

- Figure 2: I'm not sure I see what's wrong with the TMJs in this case? It might be worth explaning in you caption (lines 204-206) what the experts saw on this image.

Thank you. We added a summary of findings in the OPT shown in Figure 2 in the respective caption.

Caption Figure 2:

The left panel shows AOIs located around anomalies on an OPT. In addition to a bilateral shortening of the collum mandibulae, the colla and condyles present hypoplastic on both sides. Shortened roots of tooth 16, lacking the apical tips with periapical translucencies, possibly corresponding to a status post root resection.

In region 26, 27 and 28 spheroid, sharply defined homogeneous opacification (projecting on the maxillary sinus floor) corresponding to a mucosal antral pseudocyst.

Translucencies in the approximal areas of teeth 46 and 47 possibly indicating caries.

(Dental notation according to the FDI-system). Gridded AOIs, as displayed in the right panel were used for the computation of the overall gaze coverage.

Reviewer #3: 

The article is simple but the idea is satisfactory.

About English language: The manuscript needs an English review in order to sound as native as possible, preferably changing some words that can be understood but are not routine words in academic writing. For example, in the first phrase of the introduction, the words: "Taking radiographs" sound weird. Also, in academic writing, passive voice is preferable, and the article is almost entirely wrote in active voice.

Also, punctuation needs review.

Thank you for your suggestions. A native speaker proofread the manuscript to improve the language and punctuation. Regarding the use of active voice, we adhered to the guidelines of the American Psychological Association that recommend using “…the active rather than the passive voice” (APA Publication Manual, Sixth Edition, Section 3.18, p.77). 

# Introduction:

Please, provide a reference to the phrase: "The reason for the lack of feedback often lies in the fact that medical teachers do not have sufficient time and resources to review their students’ diagnostic competence for such a high number of radiographs." - I don't think the introduction is the right place to author's opinion.

Thank you for this comment. We agree that we should be careful with such statements in the introduction. However, with massed practice we aim at investigating an educational setting that did not get much attention from researchers over the past decades. Therefore, already in the introduction we want to explain why this particular setting is used in dentistry education. In the described scenario it is not realistic to instruct four students, let them evaluate and consecutively give feedback on 400 single panoramic radiographs within a 24 hours curriculum. This is therefore not our opinion but rather a description of the current situation in dentistry education.

I suggest you to explain further the meaning of "massed practice" with proper references. Explain how the practice is performed in a summarized and systematic explanation. Maybe you can use a figure or a flow chart to illustrate and clearer this point.

Thank you. We agree that it is eminently important to explain the meaning of “massed practice” unequivocally. Therefore, we added an explanation (see below). We also believe that Nodine and Mello-Thoms and Sowden have done outstanding work in this context and to the best of our knowledge are the only ones to address this much used learning strategy in radiology. Apart from that, the massed practice procedure was only investigated for very different tasks in general educational psychology (Radosevich & Donovan, 1999; Kornell & Bjork, 2008). Therefore, we were unable to identify any further, even more appropriate references.

Line 62

A frequently used and rather traditional instructional method for teaching students how to read and interpret radiographs is massed practice. Here students are required to provide a full written description of their observations including the identified anomalies for each radiograph. This procedure is repeated for a high number of radiographs, which are selected to reflect the full range of potential anomalies that students could be exposed to. No other learning activities are interspersed and only limited corrective feedback, if any, is provided. 

In the phrase: "According to Nodine and Mello-Thoms [6]“massed practice is the main change agent in achieving expertise” (p. 868)," please, check if the reference citation is correct. 

Thank you for pointing out this issue. We carefully checked the Vancouver style, which is the basis for the PLOS ONE style and in addition we checked published manuscripts in PLOS ONE for the correct way to indicate a direct quote in the text. However, we found very different implementations ranging from citing just the source without page number after the quote to different ways of indicating source and page number before or after the quote. Since we think it is important to make clear where exactly the quote can be found in the source, we used a style, that is actually used in published PLOS One manuscripts and is as well suggested by the Mendeley PLOS ONE style sheet. 

Line 90

According to Nodine and Mello-Thoms “massed practice is the main change agent in achieving expertise” [6] (p. 868), with…

Please, add references to the phrases: "As a consequence, it may be

particularly challenging to develop visual expertise regarding OPTs for a number of

reasons. A high interindividual variability of the visual appearance of both normal and

abnormal anatomy makes it difficult to detect anomalies. In addition, even OPTs of

apparently healthy patients can often show several anomalies"

Thank you. We added respective references:

Line 149

Consequently, it may be particularly challenging to develop visual expertise regarding OPTs for a number of reasons. A high interindividual variability of the visual appearance of both normal and abnormal anatomy as well as the phenomena caused by artifacts and superimpositions makes it difficult to detect anomalies [28–31]. In addition, even panoramic radiographs of apparently “dentally healthy patients” can often show one or more dental or non-dental anomalies [32–36].

(added) References:

28. Akkaya N, Kansu Ö, Kansu H, Çağirankaya LB, Arslan U. Comparing the accuracy of panoramic and intraoral radiography in the diagnosis of proximal caries. Dentomaxillofacial Radiol. 2006;35: 170–174. doi: 10.1259/dmfr/26750940

29. Molander B. Panoramic radiography in dental diagnostics. Swedish Dent J Suppl. 1996;119: 1–26. 

29. Nardi C, Calistri L, Grazzini G, Desideri I, Lorini C, Occhipinti M, et al. Is panoramic radiography an accurate imaging technique for the detection of endodontically treated asymptomatic apical periodontitis? J Endod. 2018;44: 1500–1508. doi: 10.1016/j.joen.2018.07.003

31. Perschbacher S. Interpretation of panoramic radiographs. Aust Dent J. 2012;57: 40–45. doi:10.1111/j.1834-7819.2011.01655.x

32. Laganà G, Venza N, Borzabadi-Farahani A, Fabi F, Danesi C, Cozza P. Dental anomalies: Prevalence and associations between them in a large sample of non-orthodontic subjects, a cross-sectional study. BMC Oral Health. 2017;17: 1–7. doi:10.1186/s12903-017-0352-y

33. Hernándes G, Plaza SP, Cifuentes D, Villalobos LM, Ruiz LM. Incidental findings in pre‐orthodontic treatment radiographs. Int Dent J. 2018;68: 320–326. doi: 10.1111/idj.12389

34. Schroder AGD, de Araujo CM, Guariza-Filho O, Flores-Mir C, de Luca Canto G, Porporatti AL. Diagnostic accuracy of panoramic radiography in the detection of calcified carotid artery atheroma: a meta-analysis. Clin Oral Investig. 2019;23: 2021–2040. doi:10.1007/s00784-019-02880-6

35. Macdonald D, Yu W. Incidental findings in a consecutive series of digital panoramic radiographs. Imaging Sci Dent. 2020;50: 53–64. doi:10.5624/ISD.2020.50.1.53

36. Monteiro IA, Ibrahim C, Albuquerque R, Donaldson N, Salazar F, Monteiro L. Assessment of carotid calcifications on digital panoramic radiographs: Retrospective analysis and review of the literature. J Stomatol Oral Maxillofac Surg. 2018;119: 102–106. doi:10.1016/j.jormas.2017.11.009

# Materials and Methods

Participants: you mentioned 55 students in the abstract, however, in the first paragraph of M&M you mentioned, after exclusion criteria applied, 47 students as your final sample. Please, correct it or clarify the numbers divergence.

Thank you. We corrected the number of the participants in the abstract.

#Discussion

Again, maybe it would be better to express your opinion in the discussion, for exemple. Remove or re-wright theses paragraphs.

Sorry, we do not know which paragraphs you were referring to. This information is missing in the comments we received.

Introduction, last two paragraphs: I don't think Introduction is the right place to express your opinion and what you expect from the study. You can express it in the discussion.

In the last two paragraphs of the introduction we give an overview of the state of research and highlight the research gap. In the last sentence, we give a summary on what our study is about. We do not think that we expressed our opinion in the mentioned paragraphs (except the reason for missing or limited corrective feedback – see our answer above). Therefore, we do not see where we could change the introduction any further without leaving out relevant research and the argumentation regarding the relevance of our present study.

I suggest you to improve your discussion with other references and comparisons with different studies, such as, for example:

* Performance evaluation of different observers in the interpretation of panoramic radiographs by the mandibular cortical index. DOI: 10.15448/1980-6523.2018.1.29202

*Observer performance in diagnosing osteoporosis by dental panoramic radiographs: results from the osteoporosis screening project in dentistry: https://doi.org/10.1016/j.bone.2008.03.014

Thank you for this suggestion. We added the studies to the discussion section to highlight why our study complements existing research, since the two suggested studies are generally interested in expert-novice differences in diagnosing part of a panoramic radiograph rather than evaluating a common educational procedure in dentistry education referring to students’ skill develoment.

Line 487

To summarize, the present study complements existing research on medical image interpretation [12–14,53–55] in that it focuses on the effects of training of students rather than on expert-novice differences and it uses OPTs rather than, for instance, chest radiographs, which require different types of visual search processes.

(added) references:

54. Taguchi A, Asano A, Ohtsuka M, Nakamoto T, Suei Y, Tsuda M, et al. Observer performance in diagnosing osteoporosis by dental panoramic radiographs: Results from the osteoporosis screening project in dentistry (OSPD). Bone. 2008;43: 209–213. doi:10.1016/j.bone.2008.03.014

55. Munhoz L, Kim JH, Park M, Aoki EM, Abdala R, Arita ES. Performance evaluation of different observers in the interpretation of panoramic radiographs by the mandibular cortical index. Rev Odonto Cienc. 2019;33: 6–10. doi:10.15448/1980-6523.2018.1.29202

---

## [Decision Letter · Decision Letter 1]

27 Oct 2020

PONE-D-20-05443R1

How massed practice improves visual expertise in reading panoramic radiographs in dental students: An eye tracking study

PLOS ONE

Dear Dr. Richter,

Thank you for submitting your manuscript to PLOS ONE. After careful consideration, we feel that it has merit but does not fully meet PLOS ONE’s publication criteria as it currently stands. Therefore, we invite you to submit a revised version of the manuscript that addresses the points raised during the second review process.

We look forward to receiving your revised manuscript.

Kind regards,

Ezio Lanza, M.D.

Academic Editor

PLOS ONE

**Comments to the Author**

Reviewer #1: Overall, the authors are to be commended for the efforts made to address the comments made by the reviewers.

Regarding my comment on the inexistence of a control group, the authors argued that “it is unclear what the control group would be”. As an example, I might mention a study conducted by Andersen et al, which managed to compare the impact of massed and distributed practice in the learning curves of virtual mastoidectomy (Andersen SA et al. JAMA Otolaryngol Head Neck Surg. 2015; 141: 913-8).

Hence, as much as I understand that it may not be suitable to change the study design at this stage, it is, nonetheless, important that the authors acknowledge the weaknesses of the current approach.

Furthermore, discussion-wise I suggest that the authors discuss the contextual interference hypothesis, which proposes that “when learning multiple skills, massing practice leads to better within-day acquisition, whereas random practice leads to better retention and transfer” (Savion-Lemieux T, Penhune VB.Exp Brain Res. 2010; 204: 271-81). In other words, the authors should discuss the potential variations of massed practice’s outcomes, depending on the task/learning objectives.

---

## [Author Response · Author response to Decision Letter 1]

11 Nov 2020

Dear Dr. Lanza, dear Reviewer, 

we would like to thank the editor and the reviewer for the assessment of our manuscript and the feedback offered by the review. In the following, we will address the comments one by one. Reviewer comments are printed in regular font, responses in italics. Moreover, we have added sections from the manuscript whenever we made a change in it (in italics and quotation marks). We hope that this way we have addressed all issues raised by the reviewer in a satisfactory manner. 

The authors 

Comments from the Editor:

Dear Dr. Richter,

Thank you for submitting your manuscript to PLOS ONE. After careful consideration, we feel that it has merit but does not fully meet PLOS ONE’s publication criteria as it currently stands. Therefore, we invite you to submit a revised version of the manuscript that addresses the points raised during the second review process.

We look forward to receiving your revised manuscript.

Kind regards,

Ezio Lanza, M.D.

Academic Editor

PLOS ONE

5. Review Comments to the Author 

Reviewer #1: 

Reviewer #1: Overall, the authors are to be commended for the efforts made to address the comments made by the reviewers.

Thank you for this overall positive assessment of our revised manuscript and your constructive feedback.

Regarding my comment on the inexistence of a control group, the authors argued that “it is unclear what the control group would be”. As an example, I might mention a study conducted by Andersen et al, which managed to compare the impact of massed and distributed practice in the learning curves of virtual mastoidectomy (Andersen SA et al. JAMA Otolaryngol Head Neck Surg. 2015; 141: 913-8).

Hence, as much as I understand that it may not be suitable to change the study design at this stage, it is, nonetheless, important that the authors acknowledge the weaknesses of the current approach.

Thank you very much for this comment. We agree that it is necessary to acknowledge the weakness of our approach regarding the missing control group. We therefore addressed this aspect in the discussion section:

l.460

Also in radiology teaching and surgical skill training, studies have shown that spaced procedures are beneficial compared to massed practice [51,52]. Future research should therefore specifically compare massed and spaced practice for learning how to diagnose OPTs to identify potentials for further improvements of students’ accuracy. The present study did not aim at contrasting different training approaches because our focus was on the effects that current practice has on students’ skill development. Therefore, we implemented the present study in a within-subjects design within students’ regular training to achieve a high ecological validity. We acknowledge that this approach has its limitations due to a lack of a control group that would have been trained with a different approach.

(added) References:

52. Andersen SAW, Konge L, Cayé-Thomasen P, Sørensen MS. Learning curves of virtual mastoidectomy in distributed and massed practice. JAMA Otolaryngol - Head Neck Surg. 2015;141: 913–918. doi:10.1001/jamaoto.2015.1563

Furthermore, discussion-wise I suggest that the authors discuss the contextual interference hypothesis, which proposes that “when learning multiple skills, massing practice leads to better within-day acquisition, whereas random practice leads to better retention and transfer” (Savion-Lemieux T, Penhune VB.Exp Brain Res. 2010; 204: 271-81). In other words, the authors should discuss the potential variations of massed practice’s outcomes, depending on the task/learning objectives.

Thank you for this suggestion. However, it is important to note that spaced vs. massed training is different from what has been proposed in research on contextual interference and interleaved practice, respectively. Spaced learning refers to the recommendation that when training a single skill, there should be pauses between multiple training sessions rather than training the same skill in a massed way. Thus, spaced vs. massed practice solely refers to the timing of the training. Contextual interference / interleaved practice refers to the training of multiple skills or skill components. Here it is recommended that these different skills are not to be trained one by one in a blocked fashion; rather, their training should be interleaved (abcabc etc.). It is true that in the case of interleaved practice this will mean that training of a single skill will be spaced at the same time. Nevertheless, the mechanism for the contextual interference effect are fundamentally different from that underlying spaced learning. In particular, interleaved practice is beneficial because it allows contrasting different training experiences, thereby highlighting that different contexts require application of different skills. The beneficial effects of interleaved practice only occur in delayed tests, whereas during training interleaved practice typically yields worse training performance. The confound between interleaved practice and spaced learning has been discussed and investigated, for instance, by Taylor and Rohrer (2009), who showed that interleaved practice has an effect above and beyond spacing. Because our study focused on spacing rather than interleaved practice, we have decided to not elaborate the concept of contextual interference in order to not cause any confusion of these two instructional principles. Rather, to highlight the focus of our paper on spaced learning, we added a footnote as follows:

l.470

Importantly, the current study refers only to the effects of massed practice regarding training of a single skill, where it has been suggested that training of that single skill should occur spaced in time. This recommendation is not to be confused with another instructional design principle, for which multiple labels have been used in the literature, that is, the contextual inference effect [53–55], the variability effect [56,57] or interleaved practice [58], respectively. Here it is suggested that when training multiple skills, these should be trained in an interleaved way (abcabcabc etc.) to highlight the differences between them. While interleaved practice of multiple skills results in spaced learning of the same skills, it is different from the focus of the present study.

(added) References:

553. Battig WF. The flexibility of human memory. In: Lermak LS, Craik FIM, editors. Levels of processing in human memory. Hillsdale, NJ: Erlbaum; 1979. pp. 23–44. 

54. Shea JB, Morgan RL. Contextual interference effects on acquisition and transfer of a complex motor task. J Exp Psychol Hum PLearning Mem. 1979;5: 179–187. 

55. Brady F. A theoretical and empirical review of the contextual interference effect and the learning of motor skills. Quest. 1998;50: 266–293. doi:10.1080/00336297.1998.10484285

56. Paas FGWC, Van Merriënboer JJG. Variability of worked examples and transfer of geometrical problem-solving skills: A cognitive-load approach. J Educ Psychol. 1994;86: 122–133. doi:10.1037//0022-0663.86.1.122

57. Likourezos V, Kalyuga S, Sweller J. The Variability Effect: When Instructional Variability Is Advantageous. Educ Psychol Rev. 2019;31: 479–497. doi:10.1007/s10648-019-09462-8

58. Taylor K, Rohrer D. The effects of interleaved practice. Appl Cogn Psychol. 2009;24: 837– 848. doi:https://doi.org/10.1002/acp.1598

---

## [Editor Report · Decision Letter 2]

16 Nov 2020

How massed practice improves visual expertise in reading panoramic radiographs in dental students: An eye tracking study

PONE-D-20-05443R2

Dear Dr. Richter,

We’re pleased to inform you that your manuscript has been judged scientifically suitable for publication and will be formally accepted for publication once it meets all outstanding technical requirements.

Kind regards,

Ezio Lanza, M.D.

Academic Editor

PLOS ONE

---

## [Editor Report · Acceptance letter]

20 Nov 2020

PONE-D-20-05443R2 

How massed practice improves visual expertise in reading panoramic radiographs in dental students: An eye tracking study 

Dear Dr. Richter:

I'm pleased to inform you that your manuscript has been deemed suitable for publication in PLOS ONE. Congratulations! Your manuscript is now with our production department. 

Kind regards, 

on behalf of

Dr. Ezio Lanza 

Academic Editor

PLOS ONE